# Selective Detection of Penicillin G Antibiotic in Milk by Molecularly Imprinted Polymer-Based Plasmonic SPR Sensor

**DOI:** 10.3390/biomimetics6040072

**Published:** 2021-12-14

**Authors:** Monireh Bakhshpour, Ilgım Göktürk, Nilay Bereli, Fatma Yılmaz, Adil Denizli

**Affiliations:** 1Department of Chemistry, Hacettepe University, Ankara 06800, Turkey; monir.b1985@gmail.com (M.B.); ilgim@hacettepe.edu.tr (I.G.); bereli@hacettepe.edu.tr (N.B.); 2Department of Chemistry Technology, Bolu Abant Izzet Baysal University, Bolu 14900, Turkey; fyilmaz71@gmail.com

**Keywords:** Ag nanoparticles, molecularly imprinted polymers, sensor, penicillin G, surface plasmon resonance

## Abstract

Molecularly imprinted polymer-based surface plasmon resonance sensor prepared using silver nanoparticles was designed for the selective recognition of Penicillin G (PEN-G) antibiotic from both aqueous solution and milk sample. PEN-G imprinted sensors (NpMIPs) SPR sensor was fabricated using poly (2-hydroxyethyl methacrylate-N-methacroyl-(L)-cysteine methyl ester)-silver nanoparticles-N-methacryloyl-L-phenylalanine methyl ester polymer by embedding silver nanoparticles (AgNPs) into the polymeric film structure. In addition, a non-imprinted (NpNIPs) SPR sensor was prepared by utilizing the same polymerization recipe without addition of the PEN-G template molecule to evaluate the imprinting effect. FTIR-ATR spectrophotometer, ellipsometer, contact angle measurements were used for the characterization of NpMIPs SPR sensors. The linear concentration range of 0.01–10 ng/mL PEN-G was studied for kinetic analyses. The augmenting effect of AgNPs used to increase the surface plasmon resonance signal response was examined using polymer-based PEN-G imprinted (MIPs) sensor without the addition of AgNPs. The antibiotic amount present in milk chosen as a real sample was measured by spiking PEN-G into the milk. According to the Scatchard, Langmuir, Freundlich and Langmuir–Freundlich adsorption models, the interaction mechanism was estimated to be compatible with the Langmuir model.

## 1. Introduction

Surface plasmon resonance (SPR) sensors are widely used to detect molecules because they are simple to prepare, inexpensive, have high specificity and sensitivity, do not need labeling and can perform real-time measurements with ease of miniaturization [1,2,3,4]. SPR is an optical technique based on the intensity of reflected light in a prism covered with a thin metal film. SPR sensors have been used widely to detect a variety of biomolecules due to their simplicity and other advantages [5]. In recent years, SPR sensors, which can measure the change in microscopic refractive index, have been increasingly used in the field of food analysis [6,7,8]. However, the development of high-throughput, highly sensitive, cost-effective methods to be used for in situ detection of food contaminants is still limited [9,10,11]. This is due to multiple signal overlapping or mutual interference and cross-reactions between different analytes with similar molecular structures [12,13].

Molecular imprinting technology creating antibody-like artificial materials with tailor-made recognition sites has become a well-established analytical tool for molecular recognition. Molecularly imprinted polymers (MIPs) with many advantages like excellent selectivity, simple, cost-effective, inexpensive preparation methods and high chemical and physical stability have become very popular when designing sensors [14,15,16,17]. MIPs are prepared by using the functional monomer, cross-linker, template, and initiator in a polymeric matrix that can function as natural antibodies. MIPs are suitable for molecular recognition, which is the reason for much of the interest in MIPs. The main advantage of MIPs is their ability to detect the target molecule with high selectivity even in the presence of other similar structures [18,19,20,21]. Therefore, MIPs are used as unique technology for the detection of biological molecules in sensors. The use of MIPs in sensors areas has been reported for the detection of pharmaceuticals [22]. The area available for binding of the target is increased by using recognition layers such as nanoparticle-based MIPs in SPR sensors; this increase the sensitivity of detection [23,24,25].

The use of nanoparticle-based MIPs in SPR sensors plays an important role in the detection of molecules such as antibiotics in real samples. The presence of antibiotic residues in food products, especially in milk and in meat, leads to serious health consequences, including increased incidence of allergic reactions. Penicillin antibiotics are most commonly used for the treatment of various bacterial infections. Therefore, the detection of penicillin residues plays a vital role in the protection of public health [26]. The traditional methods for the detection of antibiotics are based on growth inhibition of *Bacillus stearothermophilus.* Unfortunately, it takes a long time to analyse these non-sensitive tests [27]. In order to avoid any hazard, the maximum residue limits (MRLs) of antibiotics in the foods have been determined according to the European Union Regulations [28,29,30,31].

In addition to growth inhibition of *Bacillus stearothermophilus* tests, the separation methods such as chromatographic assays, capillary electrophoresis [32,33,34,35] and immunoassay methods are used for the detection of antibiotics. Because of the disadvantages of these methods, such as their being time consuming, expensive and needing pollutant solvents, new sensing technologies are being developed. The sensing systems with simple, fast and specific assay methods have become a notable technology so far. Recently several sensitive sensor technologies have been reported to detect PEN-G [36,37], especially, electrochemical biosensors have the widest application area in PEN-G biosensing [38,39].

Herein, PEN-G detection was studied using the molecular imprinting method in both the aqueous solution and milk sample by SPR method. Plasmonic properties of AgNPs were combined with molecular imprinting technology to increase the sensitivity of the SPR sensor. Selectivity studies of the NpMIP SPR sensor were performed using competing molecules, amoxicillin (AMX) and ampicillin (AMP). The imprinting efficiency of the NpMIP SPR sensor for PEN-G detection was evaluated by comparing it with the non-imprinted NpNIP SPR sensor. Here, two different functional monomers were used to obtain NpMIP SPR sensor. N-Methacryloyl-(L)-Phenylalanine Methyl Ester (MAPA) and N-methacryloyl-(L)-cysteine methyl ester (MAC) functionel monomers were used for imprinting of PEN-G and for coordination with AgNPs, respectively. The characterization studies were carried out by FTIR, ellipsometer and water contact angle (CA) measurements. Adsorption kinetics were determined by passing PEN-G solutions using 0.01–10 ng/mL concentration through the NpMIP SPR sensor and reflectance values were measured. Also, the enhancement effect of Ag nanoparticles was evaluated by passing PEN-G solution through the NpMIP SPR sensor. In addition, analyzes were carried out with real samples to investigate the effects of existing residues. The reusability studies were reported for the same PEN-G solutions, which was applied by five times consecutively.

## 2. Materials and Methods

### 2.1. Materials and Instruments

N-Methacryloyl-(L)-Phenylalanine Methyl Ester (MAPA) and N-Methacryloyl-(L)-Cysteine Methyl Ester (MAC) monomers were supplied from Nanoreg (Ankara, Turkey). 2-hydroxyethyl methacrylate (HEMA), ethyleneglycol dimethacrylate (EDMA), azoisobisbutyronitrile (AIBN), Penicillin-G (PEN-G), Amoxicillin (AMX), Ampicillin (AMP) were purchased from Merck (Darmstadt, Germany). Silver nitrate and sodium citrate used for silver nanoparticles (AgNPs) preparation were also supplied by the Merck firm (Darmstadt, Germany). SPR chips (SPR-1000-050 SPR CHIP GWC) were obtained from Genoptics (Orsay, France). UV-VIS spectrophotometer (Shimadzu UV-1601, Kyoto, Japan), FTIR-ATR spectrophotometer (Thermo Fisher Scientific, Nicolet iS10, Waltham, MA, USA), auto-nulling imaging ellipsometer (EP3-Nulling Ellipsometer, Göttingen, Germany) and contact angles Kruss DSA100 (Hamburg, Germany) were used for characterization of SPR chips. SPR imager II (GWC Technologies, Madison, WI, USA) was used during the experiments.

### 2.2. Synthesis of Silver Nanoparticles

To compare the imprinting efficiency of PEN-G imprinted NpMIPs, non-imprinted NpNIPs SPR sensors were prepared, while AgNPs-free MIPs SPR sensor was prepared to evaluate the signal enhancement effect of AgNPs. Same polymerization procedure was applied except for the addition of PEN-G and AgNPs respectively. Firstly, AgNO_3_ salt was reduced to AgNPs according to the well-known Turkevich method using sodium citrate [40]. Briefly, 1.0 × 10^−3^ M AgNO_3_ solution was heated until boiling. When boiling started, sodium citrate solution was added dropwise to the silver nitrate solution. The solution’s color turning gradually to grayish yellow indicates reduction of Ag^+^ ions, and after color change, heating was continued for another 15 min and then the solution was cooled to room temperature.

After determining the average size of AgNPs by zeta size measurement, the concentration of AgNPs was estimated [41]. The number of Ag atoms per nanoparticle (N) is calculated using Equation (1), while Equation (2) is used to calculate AgNPs concentration C. In the equations, ρ stands for Ag density (10.5 g/cm^3^), M stands for Ag atomic weight, and D stands for the diameter of AgNPs, while N stands for the total number of Ag atoms, V for the solution volume, and N_A_ for Avogadro’s number.
(1)$$N=\frac{\pi }{6}\frac{{\rho D}^{3}}{M}$$
(2)$$C=\frac{N_{\mathrm{total}}}{{\mathrm{NVN}}_{A}}$$

### 2.3. Preparation of Nanosensors

AgNPs-containing PEN-G imprinted NpMIPs SPR sensors, AgNPs-containing non-imprinted NpNIPs SPR sensors and AgNPs-free, PEN-G imprinted MIPs SPR sensors were fabricated by preparing nanofilms on the gold SPR chip surfaces according to the following procedures.

Before the preparation of nanofilm on the surface of SPR chip, the gold surface of SPR chip was cleaned in 10 mL of pure ethyl alcohol, deionized water and acidic piranha solution (3:1 H_2_SO_4_, H_2_O_2_, *v*/*v*) for 10 min, respectively and then dried in a vacuum oven (200 mmHg, 37 °C) [42]. Afterwards, 3 mM allyl mercaptan (CH_2_CHCH_2_SH) solution was dropped on the gold surface of SPR chip and incubated overnight in a fume hood to introduce allyl groups to the gold surface. For the removal of the unbound allyl mercaptan molecules, SPR chip was cleaned with ethyl alcohol, and then dried in a vacuum oven (220 mmHg, 25 °C). And then, AgNPs were mixed with MAC functional monomer (0.01 nmol:0.01 mmol) to prepare AgNPs-MAC pre-complex. The complex formation of MAC monomer with AgNPs was determined by UV-VIS spectrophotometer. Also the functional monomer MAPA and PEN-G (0.1 mmol:0.01 mmol) were reacted for 1 h to prepare the MAPAPEN-G pre-complex and EDMA (0.04 mmol) monomer was mixture was added to the pre-complexed mixture. Lastly, two prepared mixtures and 2 mg of AIBN were added together and mixed for 1 h at 25 °C [43]. To obtain SPR chips the prepared monomer mixture was poured onto allyl mercaptan modified SPR chip surfaces and the polymerization process was initiated by using a UV light and allowed to occur for 1 h on the SPR chip surface for conversion of monomer mixtures to polymeric films. Schematic illustration of the preparation of NpMIPs SPR SPR sensor is shown in Figure 1.

After the polymerization was completed the PEN-G was removed from the matrices using methanol/acetic acid mixture (80:20, *v*/*v*%) as a desorption agent. The desorption solution was applied into the SPR system by renewing it until no PEN-G absorbance at 291 nm was determined by a UV-VIS spectrophotometer. NpNIP SPR sensor was obtained using the same process except for the addition of PEN-G as a template molecule. In addition, PEN-G imprinted MIPs SPR sensor was produced by the same procedure without addition of AgNPs. Non-imprinted-NpNIPs and AgNPs-free MIP SPR sensors were used for control experiments.

### 2.4. Characterization of SPR Sensors

Size distribution and concentration of AgNPs used for the preparation of NpMIPs and NpNIPs SPR sensors were estimated by zeta sizer measurements and complex formation of AgNPs with MAC monomer was monitored by UV-VIS spectrophotometer. FTIR-ATR spectrophotometer was also used to characterize the MAC-Au and MAPAPEN-G pre-complexes formed by the reaction of MAC monomer with AgNPs and reaction of MAPA with PEN-G. Total reflection amount in the wavenumber range of 400–4000 cm^−1^ was measured.

The surface characterization of NpMIPs and NpNIP SPR sensors were evaluated by ellipsometer and CA measurements. Layer thickness determination of the SPR sensor’s surface was measured at an incident angle of 62° and a wavelength of 532 nm by using an auto-nulling imaging ellipsometer for NpMIPs and NpNIP SPR chips. The surface characterization of NpMIPs and NpNIP SPR sensors were evaluated with a water CA instrument. Using the sessile drop system, the water CA values of the SPR sensors were estimated from ten different areas of the sensor surfaces and average drop angle calculation was recorded.

### 2.5. Kinetic Analyzes of Sensors

Kinetic analyses of NpMIPs, NpNIPs and MIPs SPR sensors were obtained by SPR imager II. As an adsorption buffer, the pH 4.0 acetate solution was passed through the SPR system for 10 min at a flow rate of 0.20 mL/min to equilibrate the sensors. The equilibration buffer was passed through the SPR sensor systems for 50 s after reaching equilibrium of the systems. Then, concentrations of PEN-G solutions between 0.01 and 10 ng/mL were applied sequentially to the SPR sensors. Plasmonic responses were recorded as changes in reflectance values for each sample. After reaching the equilibrium, methanol:acetic acid solution (80:20, *v*/*v*%) as desorption solution was used at the same flow rate to desorb the target PEN-G molecule from the surface of SPR sensors. In addition, to examine the effect of residues present in the milk sample, PEN-G was added to the milk sample at a concentration of 10 ng/mL and kinetic analyses were performed with the NpMIP SPR sensor.

### 2.6. Determination of Selectivity and Imprinting Efficiency

AMX and AMP molecules were selected as competing reagents due to their similar shape and size to the template PEN-G molecule for selectivity studies. The adsorption studies applied for PEN-G detection were performed using 10 ng/mL PEN-G and competitor molecules (10 ng/mL AMX and 10 ng/mL AMP solutions). The same kinetic analysis method used for PEN-G detection was applied for the AMX and AMP molecules selected as competitor agents. For examination of the imprinting effect on the PEN-G recognition by PEN-G imprinted NpMIP SPR sensor, the non-imprinted NpNIP SPR sensor was prepared and the signal responses obtained by applying 10 ng/mL PEN-G solution prepared in 10 mM pH 4.0 acetate buffer to both SPR systems were compared. PEN-G adsorption and desorption experiments of NpMIPs and NpNIP SPR sensors were performed to determine the imprinting efficiency. To evaluate the imprinting efficiency, an imprinting factor using SPR signal intensity (ΔR) values was employed using Equation (3).
I.F = ΔR _NpMIPs_/ΔR _NpNIPs_(3)

## 3. Results and Discussion

### 3.1. Characterization of SPR Sensors

The concentrations of AgNPs were estimated using a zeta sizer instrument enabling size distribution values. It is understood from the low polydispersity index value (PDI = 0.154) that monodisperse particles with homogeneous average AgNP size distribution were obtained. The absence of AgNPs with different sizes indicates no aggregation in the system (Figure 1A). The determined size of the AgNPs was 56.61 nm. The concentration of AgNPs was calculated to be 1.62 × 10^–7^ M using the average AgNP dimension determined by zeta size measurement. The size measurement of the AgNPs is shown in Figure 1A. A plot of absorbance vs. reaction time was monitored by spectrophotometric measurement and AgNP formation was confirmed by the band monitored at 420 nm, as shown Figure 1B.

For the preparation of MAC-AgNP pre-complex AgNPs and MAC functional monomer were complexed in a ratio of 0.01 nmol:0.01 mmol for 1 h in a rotator. MAC-AgNP pre-complexation and pre-complexation of MAPA monomer chosen as a functional monomer with PEN-G was confirmed also by FTIR-ATR spectrum as shown in Appendix A. When the FTIR-ATR spectrum of the MAC monomer was evaluated, amide I and amide II bands appeared at 1452 cm^−1^ and 1523 cm^−1^, respectively. At 3347 cm^−1^ a broad -OH band was observed while the carboxylic acid (C=O) stretching band was recorded at 1726 cm^−1^, and the -SH stretching band was at 2867 cm^−1^. The shifting of -SH stretching bands to 2873 cm^−1^ in the MAC-AgNPs pre-complex implies that AgNPs coordinated to the cysteine residue. FTIR-ATR spectrum of MAPA and MAPAPEN-G pre-complexes are shown in Appendix A also. The disappearance of a band at 1532 cm^−1^ is due to the MAPA monomer, which indicates the interaction with PEN-G molecule [42].

Characterization studies of NpMIPs and NpNIP SPR sensors were evaluated by FTIR-ATR spectrum, ellipsometer and CA measurements. PEN-G molecule and AgNPs incorporated into the polymeric structure were recorded when the FTIR-ATR spectrum of NpMIPs and NpNIP SPR sensors was compared. The MAC-AgNP pre-complex formed was also successfully incorporated into the polymeric structure of the NpMIP SPR sensor. The amide I band at the 1569 cm^−1^ wavenumber in the NpNIP SPR sensor disappears in the FTIR-ATR spectrum of the NpMIP SPR sensor and it indicates that the PEN-G molecule was integrated into the structure. Figure 2 shows the FTIR-ATR spectrum of the NpMIPs and NpNIPs SPR sensors.

Ellipsometer measurements were reported as 78.12 ± 1.2 and 57.1 ± 2.2 nm, respectively, for the thickness values of the NpMIPs and NpNIP SPR sensors as shown in Figure 3A,B. The possible reason for the large difference in surface thickness values recorded by ellipsometric measurements is the distribution of AgNP sizes up to 60 nm. The surface thickness differences of the PEN-G imprinted NpMIPs and non-imprinted NpNIPs SPR sensor indicate that the PEN-G imprinting was performed successfully. From the roughness distribution through the SPR sensor chip surface it was concluded that the PEN-G was homogeneously imprinted onto the SPR sensor.

The estimated CA values for the NpMIPs and NpNIP SPR sensor chip surfaces were recorded as 91.6° ± 0.2 and 87.1° ± 0.5, as shown in Figure 3C,D. It was observed that the SPR sensor surface hydrophobicity increased because of the coordination of PEN-G to MAPA with the molecular imprinting process, and as a result the water contact angle values increased.

### 3.2. Kinetic Analyses of SPR Sensors

Optical sensors based on the principles of absorbance, fluorescence and chemiluminescence are powerful detection and analysis tools. The equilibrium and kinetic isotherm parameters for the NpMIP SPR sensor were determined and the used equations were reported as in the following Equations (Equations (4)–(8));
(4)$$\text{Equilibrium kinetic analysis}\;\frac{d\Delta R}{\mathrm{dt}}=k_{a}C\left({\Delta R}_{\max}-\Delta R\right)-k_{d}\Delta R$$
(5)$$\text{Scatchard}\;\frac{{\Delta R}_{\mathrm{ex}}}{\left[C\right]}=K_{A\;}\left({\Delta R}_{\max}-{\Delta R}_{\mathrm{eq}}\right)$$
(6)$$\text{Langmuir}\;\Delta R=\left\{\frac{{\Delta R}_{\max}\;\left[C\right]}{K_{D}}+\left[C\right]\right\}$$
(7)$$\text{Freundlich}\;\Delta R={\mathrm{DR}}_{\max}\;{\left[C\right]}^{1/n}$$
(8)$$\text{Langmuir-Freundlich}\;\Delta R=\left\{\frac{{\Delta R}_{\max}\;{\left[C\right]}^{1/n}}{K_{D}}+{\left[C\right]}^{1/n}\right\}$$

In the equations % change in reflectivity (ΔR) refers to the measured SPR signal response after binding; C is the PEN-G concentration. K_A_ and K_D_ correspond to forward and reverse equilibrium constants, respectively. The values of k_a_ and k_d_ express the kinetic rate constants of the forward and backward reaction, respectively, while 1/n is the Freundlich exponent.

In Figure 4A, ΔR values versus time for different PEN-G concentrations obtained with NpMIP SPR sensor are shown. Figure 4B shows the linearity of the increasing SPR response with increasing concentrations of PEN-G applied to the NpMIP SPR sensor. It is deduced from the high regression coefficient that the binding occurs with high affinity. To explain it in another way, PEN-G is measured with high affinity by the NpMIP SPR sensor in the concentration range of 0.01–10 ng/mL. The graphs for the equilibrium and the binding kinetic analyses were shown in Figure 4C,D. In addition, R_max_, R^2^, K_A_ and K_D_ values calculated from the lines were tabulated in Table 1. When the results were examined to explain the adsorption model, it was concluded that the mechanism on which the Langmuir adsorption model is based, is the model that best fits and PEN-G is bound to the NpMIPs SPR sensor in a monolayer form. Equilibrium binding isotherm parameters are shown in Table 2.

Sensor applications as well as chromatographic methods for PEN-G detection have increased in recent years [18,42,44,45]. Sensors consist of a biorecognition layer and a transducer element so that chemical information is transformed into a useful analytical signal. Electrochemical biosensors have the widest application area in PEN-G sensing [46]. Optical sensors have powerful detection capabilities based mainly on the principles of absorbance, chemiluminescence and fluorescence. SPR sensors are in widespread use for detecting molecules. The aim of this study is to design and develop an MIP-based SPR sensor system for selective detection of PEN-G. By using AgNPs, SPR sensor has been improved for PEN-G sensing in the complex matrix without the need of any reference material. An amino acid-based NpMIPs SPR sensor with the dual ability to form both the hydrophobic matrix and the functional group supplier in one mode was prepared and used to detect PEN-G for the first time without needing any other spacer arm or functional monomer and any extra-complicated processes such as ligand immobilization. MIP is used as an attractive functional material with imprinting cavities for specific recognition of template molecules. MAPA which is bound to PEN-G molecule with its numerous binding sites introduced by the phenylalanine part provides higher sensitivity with a lower limit of detection value concerning the amino acid-based NpMIPs SPR sensor compared to the other electrochemical sensors [38,39,47] that have also been reported in the literature. Despite several relevant electrochemical sensors having a better [48] detection limit, the NpMIPs SPR sensor showed a lower limit of detection values when compared to the other sensor platforms.

In a study, penicillinase (Pen X)-rhombus porous carbon was used as the detection element for sensitive detection of penicillin sodium. The limit of detection value was demonstrated to be 2.68 × 10^−7^ mg/mL [22]. In another study, a novel sensor based on a carbo ionic liquid electrode and TiO_2_ Nano-Particles (NPs)/Ionic Liquid (IL) (octylpyridinium iodide) was developed for the sensitive determination of PENG with 2.09 nM detection value [38]. Wang and coworkers designed a novel kind of electrochemical sensor based on magnetic mesoporous hollow carbon microspheres (MHMs) as a Pen X adsorption carrier for rapid detection of penicillin sodium, and 2.655 × 10^−7^ mg/mL detection value was reported [49]. In other studies, beta-lactam antibiotics penicillin was used as a conjugated antibody with gold nanoparticles. The Au nanoparticles synthesized from Chinese lettuce leaf extract (as reductant) were used for the colorimetric detection of penicillin. Results showed that the antibody was sensitive to 1.0 nM of the penicillin studied [50]. In our previous study, we designed an amino acid-based Au nanoparticle-based MIP SPR nanosensor for sensitive detection of PEN-G. Imprinting efficiency for this nanosensor was determined to be 7.83 by comparing it with the non-imprinted nanosensor and 0.0017 ppb limit of detection was reported [42].

In this study, the sensitivity and selectivity of the sensor were enhanced by the SPR signal augmenting effect of AgNPs and imprinting of PEN-G molecules. The NpMIP sensor has been used to detect PEN-G without the requirement of extra processing spacer arm or ligand immobilization. It was proven that PEN-G was detected selectively from its analogue molecules by the molecular imprinting method. The sensitivity of the AgNP-based MIP sensor system was reported to be greater than the AuNP-based MIP sensor system.

### 3.3. Reusability Studies of NpMIP SPR Sensor

Molecular imprinted SPR sensors are resistant to harsh environmental conditions due to their stable polymeric structures and they have reusability capacity. The stability of the NpMIPs SPR sensor can be affected by the disruption of polymeric structures in the regeneration stages. The three-dimensional structure stability of the SPR sensor can be reported by reusability studies. Therefore, the reusability studies of the NpMIPs SPR sensor system were evaluated by applying 0.01 ng/mL PEN-G solution and, as seen in Figure 5A, the sensor response was plotted as a ΔR versus time graph. PEN-G solution was passed through the SPR sensor system with five replications at 0.01 ng/mL concentration.

### 3.4. Selectivity Studies of NpMIP SPR Sensor

To prepare the sensitive NpMIPs SPR sensor, the MAC monomer containing cysteine amino acid and hydrophobic MAPA monomer as a functional phenylalanine group supplier were used. The imprinted sites having size and shape memory for selective recognition increase efficiency in the rebinding of target molecules [51]. As shown in Figure 5B, the selective detection of the NpMIPs SPR sensor was evaluated by performing competitive adsorption studies in the presence of some structural analogues such as AMX and AMP. Therefore, 10 ng/mL AMX and 10 ng/mL AMP solutions were assayed with the NpMIPs SPR sensor. The imprinting efficiency of the NpMIPs SPR sensor for 10 ng/mL PEN-G detection was evaluated by comparing it with the NpNIPs SPR sensor. In Table 3, selectivity and relative selectivity coefficients for AMX and AMP obtained by NpMIPs SPR and NpNIPs SPR sensor were reported. For PEN-G detection it was reported that NpMIPs SPR sensor was 17.93 times more selective than AMX and 7.805 times more selective than AMP antibiotics. Imprinting efficiency was estimated by calculating the imprinting factor and the reported value (IF = 14.56) implies that with the imprinting process, PEN-G was detected more selectively by NpMIPs SPR sensor than NpMIPs SPR sensor.

### 3.5. The Effect of Silver Nanoparticles on Signal Enhancement

Recently, diverse types of nanomaterials such as graphene, AuNPs, carbon nanotubes have been used to enhance SPR sensor signal response. Here, we used AgNPs to increase SPR response for PEN-G determination. In addition, we compared NpMIPs SPR sensor response with MIPs SRP sensor response prepared without using any AgNPs. The results clearly showed that the AgNPs increased the signal intensity. Therefore, a novel study showed that amino acid-based monomers modified with AgNPs were used for sensitive detection of PEN-G. In addition, the applied MIP technology played an important role in selective PEN-G recognition. The obtained ΔR values for the 10 ng/mL concentration of PEN-G solutions are shown in Figure 6. PEN-G solutions were applied to both the NpMIPs and MIP systems and the recorded responses demonstrated an enhanced signal response for the NpMIPs SPR sensor when compared with the MIPs SPR sensor.

### 3.6. The Determination of PEN-G from Milk Sample

Milk solution was selected as a real sample to show sensitive recognition of PEN-G. Milk is accepted as a healthy and comprehensive nutrient among the public. Due to dairy milk being at risk of infectious diseases, antibiotic use for these infections is essential. It is rightly stated that the concentration of drug residues in milk should be lower than the limit specified by the FAO/WHO in 1969 [52]. For the preparation milk sample at first, acetonitrile was used to coagulate and deproteinize the milk [53,54]. Then the sample was filtered using a filter paper (Whatman, 125 mm) and then the supernatant was recovered. Any suspended materials were removed by centrifugation at 10,000 rpm for 30 min with the MSE 869-Minor centrifuge device. Milk sample solution with a PEN-G antibiotic was prepared by spiking a known concentration of PEN-G and was passed through a sensor. Before passing, the sample treatment process was applied to the sample of cow milk with a known concentration of PEN-G. The ΔR value was reported as 17.12. The sensorgram for 10 ng/mL PEN-G spiked milk sample is shown in Figure 7.

## 4. Conclusions

In this study, selective and sensitive SPR sensor technology was used for PEN-G recognition by utilizing molecular imprinting technology to produce the NpMIP SPR sensor. The shift in resonant angle was recorded during the experiments by applying different concentrations of PEN-G solution to the SPR sensor. The PEN-G selectivity study of NpMIP SPR sensor was performed by using AMX and AMP chosen as competitor molecules. The results show that the prepared sensor was 17.93 times more selective than AMX and 7.805 times more selective than AMP antibiotics. The calculated imprinting factor which was reported as IF = 14.56 implied that the imprinting process was performed successfully and also PEN-G was detected more selectively by the NpMIP SPR sensor than the NpNIP SPR sensor. In this study, sensitive sensor technology using AgNPs to amplify the SPR sensor response system for detection of PEN-G molecules was designed. The MIP SPR sensor was used for control experiments to evaluate the enhancing effect of AgNPs. The addition of AgNPs increased the sensitivity of the system eightfold. This sensing system was also successfully used for PEN-G recognition in milk samples. The PEN-G spiked sample at 10 ng/mL concentration was applied to the NpMIP SPR sensor and detected precisely with the molecularly imprinted SPR system. According to the Scatchard, Langmuir, Freundlich and Langmuir–Freundlich adsorption models, the interaction mechanism was estimated to be compatible with the Langmuir model. As as result, we developed simple, sensitive and non-toxic sensing technology for label-free determination of PEN-G without using any complicated coupling processes.

## Data Availability

The data used to support the findings of this study are included within the article and in the Appendix A.

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
