# Peer review of "Selective Detection of Penicillin G Antibiotic in Milk by Molecularly Imprinted Polymer-Based Plasmonic SPR Sensor"

_biomimetics, 2021, doi:10.3390/biomimetics6040072_

Round 1
Reviewer 1 Report
The paper describes the development of a penicillin sensor based on molecularly-imprinted polymers on SPR substrates. The sensor works well in comparison to non-imprinted analogues and is selective against similar distractant molecules. There are a few comments I can make that may help improve the clarity of the paper:
1) The English language needs to be improved, particularly in the introduction where it can be quite difficult to parse at times (e.g. the paragraph beginning line 38).
2) Line 119: "attaching nanofilms" - it is not clear that this is what is described in the following paragraph.
3) Line 205: a reference should be included to support the statement that the AgNPs have coordinated to the cysteine residue.
4) Lines 239-242 should really be in the introduction.
5) Finally, it wasn't clear to me until reaching section 3.7 that the "real" milk samples were processed extensively by e.g. coagulation, filtering, centrifuged etc. This is quite important, since the impression is given that this sensor would work by being placed directly in a sample of milk straight from the bottle.
Author Response
The paper describes the development of a penicillin sensor based on molecularly-imprinted polymers on SPR substrates. The sensor works well in comparison to non-imprinted analogues and is selective against similar distractant molecules. There are a few comments I can make that may help improve the clarity of the paper:
1) The English language needs to be improved, particularly in the introduction where it can be quite difficult to parse at times (e.g. the paragraph beginning line 38).
Answer: Thank you for your kind interest. This part was now revised and showed with red color. The English language of manuscript was checked and revised.
2) Line 119: "attaching nanofilms" - it is not clear that this is what is described in the following paragraph.
Answer: Thank you. This part was revised.
3) Line 205: a reference should be included to support the statement that the AgNPs have coordinated to the cysteine residue.
Answer: Thank you for your suggestion, the reference was added.
4) Lines 239-242 should really be in the introduction.
Answer: This part was added in the introduction.
5) Finally, it wasn't clear to me until reaching section 3.7 that the "real" milk samples were processed extensively by e.g. coagulation, filtering, centrifuged etc. This is quite important, since the impression is given that this sensor would work by being placed directly in a sample of milk straight from the bottle.
Answer: During the experiment, this procedure was applied to defat, coagulate and deproteinize. The method employed by Moors and Massart was adopted with little modifications. Below are references to support our work.
- https://doi.org/10.1002/slct.202103058
- Agric. Food Chem. 2004, 52, 2791−2796
- https://doi.org/10.1016/0731-7085(91)80136-W
- https://doi.org/10.3390/s16010056
Reviewer 2 Report
In this research, the selective detection method of penicillin G antibiotic in milk by molecularly imprinted polymer-based plasmonic SPR sensor was developed.
The manuscript is interesting and rather well organized. It is in the scope of the journal, therefore it can be published after some improvements and corrections:
Questions and comments for the article:
Commment about introduction:
It should be reviewed more articles related to MIP applications, advantages, etc. The articles related to the topic of this study and published by your scientific group should be included also. It should be clearly and correctly indicated what was done in their previous studies and what is new in the current research.
Lines 11-12: “poly(2-hydroxyethyl methacrylate-N-methacroyl-(L)-cysteine methyl ester-silver nanoparticles-N-methacryloyl-L-phenylalanine methyl ester polymer” – a bracket is missing.
Line 58: “Unfortunately” should be written instead of “Uunfortunately”.
Line 78: you should add the abbreviation: contact angle (CA).
Comment for methodology: Could you make a sensor preparation scheme? Check the methodology carefully. The current description of the methodology is not detailed enough.
Line 117: The description is not clear. It should be rewritten. AgNPs-free (MIPs) - what does it mean?
Line 121: the sentence “AgNPs were synthesized according to the well-known Turkevich method [18]” should be deleted. It was repeated two times.
Line 124: the description of the procedure of the attachment of allyl groups onto the gold surface is not detailed enough. You should add information about concentration, duration of the incubation, solvent, volume, etc. or if you used the procedure from literature you should add the reference.
Line 126: check the concentrations of AgNPs and MAC. In line 196 it is written different concentrations.
Line 131: For chemically induced polymerization it is very important to control the concentration and temperature because reaction rate/final structure (pore size, thickness of walls, density, etc.) is really very dependent on them. Please add the temperature of the reaction.
Lines 134-136: it should be described more accurately how the extraction of template molecule was performed: did you soak the whole sensor into the methanol/acetic acid mixture? What volume is used each time? How long you were soaking? Or you were washing in an SPR cell?
Line 183: The equation should be written in a correct way.
Fig. 3: Quality of figures 3A and B should be improved. On the axis Z, there is too much empty space.
Line 244: Equations 3-7: the description of equations should be improved.
Figure 5, line 289: what is demonstrated in the figure: reusability or repeatability of analysis?
Table 3: check the formatting of the figure.
Line 314: I don’t understand the sentence.
Line 322: You should add discussions and compare the obtained results with similar studies. At least obtained in your group. Which NP does the sensitivity of the system increase at a higher level AuNP or AgNP? To which template (penicillin or amoxicillin) the sensor is more suitable in similar conditions?
Line 331: for indication of IF use “=” instead of “:”
Line 347: Author contribution and funding should be added.
Comments about references:
The list of references seems too short. In my opinion, it should be extended.
I think you could find such articles:
1) some articles with AIBN initiation polymerization should be cited.
2) the articles with the application of chromatographic methods could be replaced with SPR and ellipsometry application references.
3) you could add some more articles with MIP applications for the determination of antibiotics.
the group of Denizli has several articles with MIP with antibiotic imprints:
- Faalnouri, S., Çimen, D., Bereli, N., & Denizli, A. (2020). Surface Plasmon Resonance Nanosensors for Detecting Amoxicillin in Milk Samples with Amoxicillin Imprinted Poly (hydroxyethyl methacrylate‐N‐methacryloyl‐(L)‐glutamic acid). ChemistrySelect, 5(15), 4761-4769.
- Development of Molecularly Imprinted Polymer‐Based Optical Sensor for the Sensitive Penicillin G Detection in Milk.
they should be added to the list of references.
Here are some studies and review articles I recommend:
- Ramanavicius, A. Ramanavicius. Conducting Polymers in the Design of Biosensors and Biofuel Cells. Polymers 2021, 13, 49.
- Ramanavicius, A. Jagminas, A. Ramanavicius, Advances in molecularly imprinted polymers based affinity sensors (Review). Polymers 2021, 13, 974.
- Jamieson, F. Mecozzi, R.D. Crapnell, W. Battell, A. Hudson, K. Novakovic, A. Sachdeva, F. Canfarotta, C. Herdes, C.E. Banks, H. Snyder, M. Peeters, Approaches to the Rational Design of Molecularly Imprinted Polymers Developed for the Selective Extraction or Detection of Antibiotics in Environmental and Food Samples, Physica status solidi (a), 218 (2021) 2100021.
- A. Lorenzo, A.M. Carro, C. Alvarez-Lorenzo, A. Concheiro, To Remove or Not to Remove? The Challenge of Extracting the Template to Make the Cavities Available in Molecularly Imprinted Polymers (MIPs), International Journal of Molecular Sciences, 12 (2011) 4327-4347.
- Nagraik, A. Sharma, D. Kumar, P. Chawla, A.P. Kumar, Milk adulterant detection: Conventional and biosensor based approaches: A review, Sensing and Bio-Sensing Research, 33 (2021) 100433.
Author Response
- Comments and Suggestions for Authors
In this research, the selective detection method of penicillin G antibiotic in milk by molecularly imprinted polymer-based plasmonic SPR sensor was developed. The manuscript is interesting and rather well organized. It is in the scope of the journal, therefore it can be published after some improvements and corrections:
Questions and comments for the article:
Commment about introduction:
It should be reviewed more articles related to MIP applications, advantages, etc. The articles related to the topic of this study and published by your scientific group should be included also. It should be clearly and correctly indicated what was done in their previous studies and what is new in the current research.
Answer: Thank you for your suggestion, the references were added to show the main aim of this study and highlighted the novelty of this research against the previous studies.
Lines 11-12: “poly(2-hydroxyethyl methacrylate-N-methacroyl-(L)-cysteine methyl ester-silver nanoparticles-N-methacryloyl-L-phenylalanine methyl ester polymer” – a bracket is missing.
Line 58: “Unfortunately” should be written instead of “Uunfortunately”.
Answer: Thank you. We corrected these mistakes in the manuscript.
Line 78: you should add the abbreviation: contact angle (CA).
Answer: Thank you for your suggestions. The abbreviation of contact angle was added in the manuscript.
Comment for methodology: Could you make a sensor preparation scheme? Check the methodology carefully. The current description of the methodology is not detailed enough.
Answer: The schematic preparation of the sensor was added. In addition, the detail of methodology was added.
Schematic 1. The schematic illustration of the preparation of Ag nanoparticles based PEN-G imprinted SPR sensor.
Line 117: The description is not clear. It should be rewritten. AgNPs-free (MIPs) - what does it mean?
Answer: This section was revised ‘‘PEN-G imprinted with using AgNPs (NpMIPs), PEN-G non-imprinted with using AgNPs (NpNIPs) and PEN-G imprinted without using AgNPs (MIPs) SPR sensors were prepared by preparing nanofilms on the gold SPR chip surfaces.’’
Line 121: the sentence “AgNPs were synthesized according to the well-known Turkevich method [18]” should be deleted. It was repeated two times.
Answer: Thank you. We deleted the repeated sentence.
Line 124: the description of the procedure of the attachment of allyl groups onto the gold surface is not detailed enough. You should add information about concentration, duration of the incubation, solvent, volume, etc. or if you used the procedure from literature you should add the reference.
Answer: Thank you. The detail was added. Before the preparation of nanofilm on the surface of SPR chip, the gold surface of SPR chip was cleaned in 10 mL of pure ethyl alcohol, deionized water and acidic piranha solution (3:1 H2SO4, H2O2, v/v) for 10 min, respectively and then dried in a vacuum oven (200 mmHg, 37 °C). Afterwards, 3 mM allyl mercaptan (CH2CHCH2SH) solution was dropped on the gold surface of SPR chip and incubated overnight in a fume hood to introduce allyl groups to the gold surface. For the removal of the unbound allyl mercaptan molecules, SPR chip was cleaned with ethyl alcohol, and then dried in a vacuum oven (220 mmHg, 25 °C).
Line 126: check the concentrations of AgNPs and MAC. In line 196 it is written different concentrations.
Answer: The ‘‘(0.01 nmol:0.01 mmol)’’ is the concentration of the AgNPs and MAC. This was corrected in the manuscript.
Line 131: For chemically induced polymerization it is very important to control the concentration and temperature because reaction rate/final structure (pore size, thickness of walls, density, etc.) is really very dependent on them. Please add the temperature of the reaction.
Answer: Thank you for your kind interest. The temperature of the reaction was added.
Lines 134-136: it should be described more accurately how the extraction of template molecule was performed: did you soak the whole sensor into the methanol/acetic acid mixture? What volume is used each time? How long you were soaking? Or you were washing in an SPR cell?
Answer: The methanol/acetic acid mixture (80:20, v/v%) as a desorption agent was used for removing the target molecule. The desorption solution was applied in the SPR system for 10 min. After that, the sample was collected and determined by a UV-VIS spectrophotometer. After 10 min, no PEN-G absorbance was detected at 291 nm.
Line 183: The equation should be written in a correct way.
Answer: The equation was rewritten.
Fig. 3: Quality of figures 3A and B should be improved. On the axis Z, there is too much empty space.
Answer: Thank you for your suggestions; but the device has a software problem; so, we could not scale up the Z axis.
Line 244: Equations 3-7: the description of equations should be improved.
Answer: Thank you for your suggestions. The detail was added.
Figure 5, line 289: what is demonstrated in the figure: reusability or repeatability of analysis? check the formatting of the figure.
Answer: The reusability response of NpMIPs SPR sensor was demonstrated in Figure 5A. PEN-G solution was passed through the SPR sensor system with five replication of same concentration. In addition, the format of the figure was checked.
Line 314: I don’t understand the sentence.
Answer: The sentence was rewritten. Milk sample solution having a PEN-G antibiotic was prepared by spiking a known concentration of PEN-G and was passed through a sensor. Before passing, the sample treatment process was applied to the sample of cow milk with a known concentration of PEN-G. The ΔR value was reported as 17.12.
Line 322: You should add discussions and compare the obtained results with similar studies. At least obtained in your group. Which NP does the sensitivity of the system increase at a higher level AuNP or AgNP? To which template (penicillin or amoxicillin) the sensor is more suitable in similar conditions?
Answer: Thank you, the manuscript was revised as your suggestions.
Line 331: for indication of IF use “=” instead of “:”
Answer: The equation was rewritten.
Line 347: Author contribution and funding should be added
Answer: The author contribution and funding were added.
Comments about references:
The list of references seems too short. In my opinion, it should be extended.
I think you could find such articles:
1) some articles with AIBN initiation polymerization should be cited.
2) the articles with the application of chromatographic methods could be replaced with SPR and ellipsometry application references.
3) you could add some more articles with MIP applications for the determination of antibiotics. the group of Denizli has several articles with MIP with antibiotic imprints:
- Faalnouri, S., Çimen, D., Bereli, N., & Denizli, A. (2020). Surface Plasmon Resonance Nanosensors for Detecting Amoxicillin in Milk Samples with Amoxicillin Imprinted Poly (hydroxyethyl methacrylate‐N‐methacryloyl‐(L)‐glutamic acid). ChemistrySelect, 5(15), 4761-4769.
- Development of Molecularly Imprinted Polymer‐Based Optical Sensor for the Sensitive Penicillin G Detection in Milk.
they should be added to the list of references. Here are some studies and review articles I recommend:
- Ramanavicius, A. Ramanavicius. Conducting Polymers in the Design of Biosensors and Biofuel Cells. Polymers 2021, 13, 49.
- Ramanavicius, A. Jagminas, A. Ramanavicius, Advances in molecularly imprinted polymers based affinity sensors (Review). Polymers 2021, 13, 974.
- Jamieson, F. Mecozzi, R.D. Crapnell, W. Battell, A. Hudson, K. Novakovic, A. Sachdeva, F. Canfarotta, C. Herdes, C.E. Banks, H. Snyder, M. Peeters, Approaches to the Rational Design of Molecularly Imprinted Polymers Developed for the Selective Extraction or Detection of Antibiotics in Environmental and Food Samples, Physica status solidi (a), 218 (2021) 2100021.
- Lorenzo, A.M. Carro, C. Alvarez-Lorenzo, A. Concheiro, To Remove or Not to Remove? The Challenge of Extracting the Template to Make the Cavities Available in Molecularly Imprinted Polymers (MIPs), International Journal of Molecular Sciences, 12 (2011) 4327-4347.
- Nagraik, A. Sharma, D. Kumar, P. Chawla, A.P. Kumar, Milk adulterant detection: Conventional and biosensor based approaches: A review, Sensing and Bio-Sensing Research, 33 (2021) 100433.
Answer: Thank you for your interest and suggestions. We added these references in the manuscript.
Reviewer 3 Report
Review: biomimetics-1484235.
Title: Selective detection of penicillin G antibiotic in milk by molecularly imprinted polymer-based plasmonic SPR sensor.
In this research type manuscript, Authors have presented the results of studies aimed to selective detection of penicillin G in milk by molecularly imprinted polymer (MIP)-based surface plasmon resonance (SPR) sensor. The manuscript is properly prepared and the topic is of interest of the broader spectrum of scientist. However, a few problems should be addressed by Authors before further proceeding of the manuscript, as below:
- In the Introduction Section, the description in lines 38-44 suffers from lack of references. Here, recent review papers related to MIPs synthesis, characterization and properties should be disclosed (see: Chem Rev. 2019, 119, 1, 94, Materials 2021, 14, 1850, Chemosensors 2021, 9, 123, Polymers 2021, 13, 974). The discussion in lines 50-55 could be supported by references related to MIPs as parts of SPR sensors (see: Sensors 2016, 16, 1381, Trends Biotechnol. 2019, 37, 294) as well as the text in lines 55-60 related to the analysis of antibiotics in food samples (see: Food Control 2020, 118, 107381, Trends Anal. Chem. 2020, 127, 115883, Food Chem. Toxicol. 2019, 125, 462).
- In the Result and Discussion Section, Authors should discuss the mechanism of specific sorption of penicillin G on the polymeric layer that possess residues from monomers used in the synthesis, viz. N-methacryloyl-L-phenylalanine methyl ester and N-methacryloyl-L-cysteine methyl ester as well as 2-hydroxyethylmethacrylate which could also interact with analyte. It is interesting in the context of other papers, describing the use of similar monomer systems but dedicated to analysis of different analytes (see: Biosensors 2021, 11, 21).
- In the Experimental Section, the characterization (composition) of gold chip shall be revealed. It is known that the stability of gold thin layers is limited and chromium is often deposited to prevent deterioration and to improve adherence of sputtered gold layer but chromium could affect the surface plasmon resonance sensing.
The minor editorial errors should be corrected such as doubled text devoted to Turkevich method (lines 104 and 121). In my opinion, Authors shall consider to use ng/L units instead of ppb. The chemical structures of analyzed antibiotics will be helpful.
Based on above, I recommend major revision of the manuscript.
Author Response
- Comments and Suggestions for Authors
Review: biomimetics-1484235.
Title: Selective detection of penicillin G antibiotic in milk by molecularly imprinted polymer-based plasmonic SPR sensor.
In this research type manuscript, Authors have presented the results of studies aimed to selective detection of penicillin G in milk by molecularly imprinted polymer (MIP)-based surface plasmon resonance (SPR) sensor. The manuscript is properly prepared and the topic is of interest of the broader spectrum of scientist. However, a few problems should be addressed by Authors before further proceeding of the manuscript, as below:
- In the Introduction Section, the description in lines 38-44 suffers from lack of references. Here, recent review papers related to MIPs synthesis, characterization and properties should be disclosed (see: Chem Rev. 2019, 119, 1, 94, Materials 2021, 14, 1850, Chemosensors 2021, 9, 123, Polymers 2021, 13, 974). The discussion in lines 50-55 could be supported by references related to MIPs as parts of SPR sensors (see: Sensors 2016, 16, 1381, Trends Biotechnol. 2019, 37, 294) as well as the text in lines 55-60 related to the analysis of antibiotics in food samples (see: Food Control 2020, 118, 107381, Trends Anal. Chem. 2020, 127, 115883, Food Chem. Toxicol. 2019, 125, 462).
Answer: Thank you for your interest and suggestions. We added these references in the manuscript.
- In the Result and Discussion Section, Authors should discuss the mechanism of specific sorption of penicillin G on the polymeric layer that possess residues from monomers used in the synthesis, viz. N-methacryloyl-L-phenylalanine methyl ester and N-methacryloyl-L-cysteine methyl ester as well as 2-hydroxyethylmethacrylate which could also interact with analyte. It is interesting in the context of other papers, describing the use of similar monomer systems but dedicated to analysis of different analytes (see: Biosensors 2021, 11, 21).
Answer: Thank you, in the result and discussion section, the details of the mechanism and the important and properties of this method were discussed.
- In the Experimental Section, the characterization (composition) of gold chip shall be revealed. It is known that the stability of gold thin layers is limited and chromium is often deposited to prevent deterioration and to improve adherence of sputtered gold layer but chromium could affect the surface plasmon resonance sensing.
Answer: In this study, SPR bare gold chips (SPRchipTM, Masidon, WI, USA) were supplied for the SPRimager II instrument by GWC Technologies (Masidon, WI, USA). The thickness of the gold is 50 nm. In during the experiment, we did not any damege on the gold surface of the SPR chip. In addition, due to the surface modification, the stability of the gold and polymeric layer is increased before the preparation of nanoparticles based nanofilm on the surface of chip.
The minor editorial errors should be corrected such as doubled text devoted to Turkevich method (lines 104 and 121). In my opinion, Authors shall consider to use ng/L units instead of ppb. The chemical structures of analyzed antibiotics will be helpful.
Answer: Thank you. We deleted the repeated sentence. We decided to use ng/mL units.
Round 2
Reviewer 3 Report
Review: biomimetics-1484235-R1.
Title: Selective detection of penicillin G antibiotic in milk by molecularly imprinted polymer-based plasmonic SPR sensor.
In this revised manuscript, Authors have made corrections according to referee comments. In my opinion, the manuscript in current form could be considered for acceptance.
Please check only the correctness of the sentence in lines 48-49.